# Scaling of the Disorder Operator at Deconfined Quantum Criticality

Yan-Cheng Wang[1], Nvsen Ma[2], Meng Cheng[3, *], Zi Yang Meng[4, †],

**1** Beihang Hangzhou Innovation Institute Yuhang, Hangzhou 310023, China
**2** School of Physics, Key Laboratory of Micro-Nano Measurement-Manipulation and Physics, Beihang University, Beijing 100191, China
**3** Department of Physics, Yale University, New Haven, CT 06520-8120, U.S.A
**4** Department of Physics and HKU-UCAS Joint Institute of Theoretical and Computational Physics, The University of Hong Kong, Pokfulam Road, Hong Kong SAR, China
* m.cheng@yale.edu; † zymeng@hku.hk

June 7, 2022

## Abstract

We study scaling behavior of the disorder parameter, defined as the expectation value of a symmetry transformation applied to a finite region, at the deconfined quantum critical point in $(2{+}1)d$ in the $J$-$Q_3$ model via large-scale quantum Monte Carlo simulations. We show that the disorder parameter for U(1) spin rotation symmetry exhibits perimeter scaling with a logarithmic correction associated with sharp corners of the region, as generally expected for a conformally-invariant critical point. However, for large rotation angle the universal coefficient of the logarithmic corner correction becomes negative, which is not allowed in any unitary conformal field theory. We also extract the current central charge from the small rotation angle scaling, whose value is much smaller than that of the free theory.

# 1  Introduction

Deconfined quantum criticality (DQC) [1–3], a continuous quantum phase transition between two seemingly unrelated symmetry-breaking states, is arguably the complex behavior of novel quantum critical point beyond the paradigm of Landau-Ginzburg-Wilson. Such a transition, if exists, is believed to host a number of unusual phenomena, such as an emergent symmetry unifying order parameters of completely different microscopic origins and the presence of fractionalized spinons, among others [3–5]. Theoretically the proposed low-energy theory is a gauge theory in the strong coupling regime, posing significant challenges to analytical treatment [2, 6]. Numerical investigations of lattice models realizing such transitions have been indispensable in pushing forward our understanding of DQC from many different angles: the two-length-scale scaling as an attempt to reconcile the anomalous finite-size scaling behavior of the J-Q model [7], conserved current exploited to exhibit the emergent continuous symmetry [8], fractionalization revealed from dynamic spin spectra [3], to name a few. There has also been exiciting progress in possible experimental realization of the DQC from the pressure-driven phase transition in the Shastry-Sutherland quantum magnet $SrCu_2(BO_3)_2$ [9–11] and its theoretical implications [12–14]. The communities of quantum phase transitions, quantum magnetism and even high-energy physics, have benefited a lot from these pursuits over the years. However, the very nature of the transition itself, and basic questions such as whether the transition is continuous or not, whether the transition follows conformal invariance and accquires a proper conformal field theory (CFT) description [6, 15–25], etc, are actually still open despite the active investigations mentioned above.

In recent years, the importance of using extended operators, such as symmetry domain walls or field lines of emergent gauge field, to probe and characterize phases and phase transitions has become increasingly clear [26–30]. In particular, many exotic gapped phases can be understood in terms of the condensation of certain extended objects, spontaneously breaking the so-called higher-form symmetry. These new insights bring intriguing connections between the Landau-Ginzburg-Wilson paradigm of spontaneous symmetry breaking and more exotic phenomena of topological order [31]. Inspired by such progress, recent works have started to explore more quantitative aspects of disorder operators, which are defined as a symmetry transformation restricted to a finite region of the system, especially at quantum criticality. Ref. [32] computed the Ising disorder operator, which serves as the order parameter of a $\mathbb{Z}_2$ 1-form symmetry, by quantum Monte Carlo (QMC) simulation at the $(2+1)d$ Ising transition. The U(1) disorder operator at the $(2+1)d$ XY transition is measured in QMC simulation as well [33]. New universal scaling behavior for such disorder operators at these conformally-invariant quantum critical points (QCP) are identified [32–36]. Building upon the methodology for the computation and analysis of disorder operator established by studying conventional symmetry-breaking transitions, in this work we take this new set of tools to study the deconfined quantum criticality.

An important difference between the DQC and other QCPs studied so far in this context is that one side of the DQC exhibits valence bond solid (VBS) order, spontaneously breaking the lattice symmetry. To understand how the behavior of the disorder operator is affected by lattice symmetry breaking, we first study two different microscopic realizations of the (2+1) O(3) QCP, the bilayer and $J_1$-$J_2$ Heisenberg antiferromagnets on the square lattice, using ubiased Stochastic series expansion (SSE) [37] QMC simulations. We find the disorder operators for U(1)$_{S_z}$ symmetry obey the expected perimeter law scaling with a multiplicative logarithmic correction at the QCPs, in agreement with the prediction of the O(3) CFT. However for the $J_1$-$J_2$ model with explicit translation symmetry breaking, it is crucial to construct disorder operators only on regions whose boundary avoids the "strong" singlet bonds, in order to obtain converged results in the finite-size analysis.

With this knowledge, we proceed with the similar measurement of the U(1)$_{S_z}$ disorder operator in the $J$-$Q_3$ model of DQC, at the critical point between the Néel and VBS phases [4]. To mitigate finite-size error due to the VBS fluctuations, in our QMC measurement of the disorder operator we adjust the region according to the profile of the instantaneous VBS order. Our data reveal that although the disorder operator still obeys the scaling behavior expected for a general CFT, the universal coefficient in the logarithmic correction term becomes negative for U(1) rotation angle close to $\pi$, which we argue is incompatible with any unitary CFT and in fact suggests a large violation of unitarity. We also extract the current central charge from the small angle scaling, whose value is significantly smaller than conventional O($n$) CFT.

## 2 Three lattice models

We simulate the following three lattice models hosting the target QCPs. The first is the bilayer square lattice antiferromagnet with Hamiltonian

$$H_{\text{bilayer}} = J_1 \sum_{\langle ij \rangle} (\mathbf{S}_{1,i} \cdot \mathbf{S}_{1,j} + \mathbf{S}_{2,i} \cdot \mathbf{S}_{2,j}) + J_2 \sum_i \mathbf{S}_{1,i} \cdot \mathbf{S}_{2,i}, \qquad (1)$$

where $\mathbf{S}_{\alpha,i}$ is a spin-1/2 at site $i$ of layer $\alpha(= 1, 2)$, $\langle ij \rangle$ denotes the neareast-neighbor antiferromagnetic coupling on the square lattice. $J_2$ is the interlayer antiferromagnetic interaction. The model is illustrated in Fig. 1(a). The critical point $(J_2/J_1)_c = 2.5220(1)$ [38, 39] separating the Néel state and the symmetric product state of inter-layer singlets, belongs to the $(2+1)d$ O(3) universality class.

The next model is the square lattice $J_1$-$J_2$ Heisenberg, shown in Fig. 1(b). The Hamiltonian reads

$$H_{J_1-J_2} = J_1 \sum_{\langle ij \rangle} \mathbf{S}_i \cdot \mathbf{S}_j + J_2 \sum_{\langle ij \rangle'} \mathbf{S}_i \cdot \mathbf{S}_j, \qquad (2)$$

where $\langle ij \rangle$ denotes the thin $J_1$ bond and $\langle ij \rangle'$ denotes the thick $J_2$ bond, and the QCP $(J_2/J_1)_c = 1.90951(1)$ [40] is also known to fall within the $(2+1)d$ O(3) universality class. The reason that we study both Eqs. (1) and (2) is that although the QCPs are in the same universality class, the presence of strong $J_2$ and weak $J_1$ bonds in Eq. (2) breaks the lattice translation symmetry while Eq. (1) is fully translation-invariant. As we show below, because of the translation symmetry breaking, the domain $M$ must be chosen so that its boundary avoids strong singlet bonds to correctly extract the scaling behavior of the disorder operator.

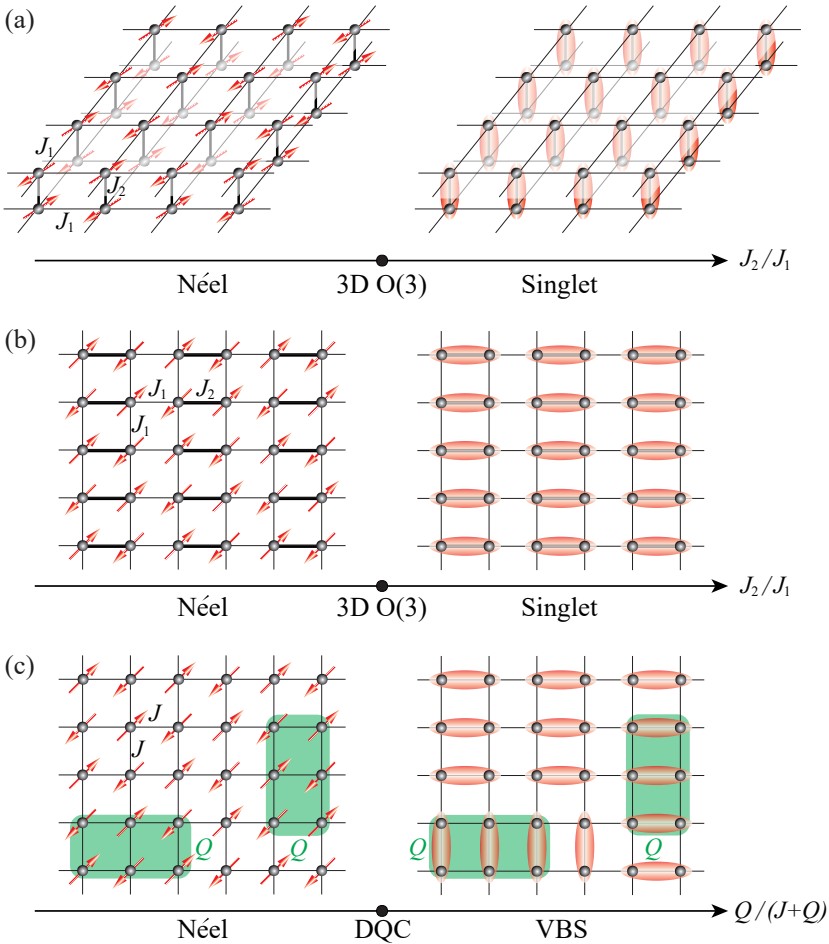

Figure 1: The three lattice models: (a) the bilayer square lattice antiferromagnetic Heisenberg model, (b) the square lattice $J_1$-$J_2$ antiferromagnetic Heisenberg model and (c) the $J$-$Q_3$ model. (a), (b) exhibits $(2+1)d$ O(3) QCP as $J_2/J_1$ is tuned [38–40] and (c) gives rise to DQC [4].

The last model is the $J$-$Q_3$ model as illustrated in Fig. 1(c) with the following Hamiltonian,

$$H_{J-Q_3} = -J \sum_{\langle ij \rangle} P_{ij} - Q \sum_{\langle ijklmn \rangle} P_{ij} P_{kl} P_{mn}. \tag{3}$$

Here $P_{ij} = \frac{1}{4} - \mathbf{S}_i \cdot \mathbf{S}_j$ is the two-spin singlet projector. The quantum critical point separating the Néel and VBS states is at $[Q/(J+Q)]_c = 0.59864(5)$ [4,41] (see Appendix B for details [42]). While the VBS order vanishes at the QCP after extrapolating to the thermodynamic limit, in a finite system there is always a small but non-zero VBS order. Therefore the computation of the disorder parameter may suffer from similar kinds of lattice effect that occurs in the $J_1$-$J_2$ model. To eliminate such effect as much as possible, in our QMC measurement of the disorder operator we adjust the region according to the profile of the instantaneous VBS order to achieve robustly converged results from finite-size analysis (see Appendix B for details).

## 3 Disorder operator

All three lattice models have SU(2) spin rotational symmetry. For any U(1) subgroup we will define a disorder operator that depends on the U(1) rotation angle. Without loss of generality, we will consider spin rotations around the $z$ and the U(1) symmetry transformations are implemented by $U(\theta) = \prod_i e^{i\theta(S_i^z - \frac{1}{2})}$, where $S_i^z$ is the U(1) charge on site $i$. For a region $M$, we define the disorder operator $X_M(\theta) = \prod_{i \in M} e^{i\theta(S_i^z - \frac{1}{2})}$. The ground state expectation value $\langle X_M(\theta) \rangle$ will be referred to as the disorder parameter. The scaling behavior of $\langle X_M(\theta) \rangle$ in various phases, especially the dependence on the geometry of $M$, has been studied thoroughly in [33]. In a U(1)-symmetric phase, such as the singlet ground state in $H_{\text{bilayer}}$ and $H_{J_1-J_2}$ models, $\langle X_M(\theta) \rangle$ is expected to obey a perimeter law $|\langle X_M(\theta) \rangle| \sim e^{-a_1(\theta)l}$, where $l$ is the perimeter of the region $M$. In the ordered (U(1) symmetry breaking) phases, such as the Néel phase of the three models, it was found that $|\langle X_M(\theta) \rangle| \sim e^{-b(\theta)l \ln l}$ [33,43]. Our focus in this work, however, is the disorder operator at QCPs in Fig. 1, in particular that of the DQC. Previous studies of the $(2+1)d$ Ising and O(2) transitions, as well as other gapless critical theories [32–34,36] suggest that for large $l$, $\ln|\langle X_M(\theta) \rangle|$ takes the following general form for a rectangle region:

$$\ln |\langle X_M(\theta) \rangle| = -a_1 l + s \ln l + a_0. \tag{4}$$

Here all the coefficients are functions of $\theta$. We note that as an expansion in large $l$, Eq. (4) contains all terms compatible with scale invariance (dropping those that decay with $l$). The universal logarithmic correction, which translates into a power law $l^s$ in $|\langle X_M \rangle|$, originates from sharp corners of the region. In general $s$ is a universal function of both $\theta$ and the opening angle(s) of the corners (all $\pi/2$ in this case) [33,35]. Similar corner contributions were known to arise for Rényi entropy in a CFT [44,45], which can be understood as the disorder operator of the replica symmetry. In Ref. [33], analytical arguments were presented to support the universal corner correction for the disorder operator and the universal coefficient $s$ is found to be given by $s(\theta) \approx \frac{C_J}{(4\pi)^2} \theta^2$ as $\theta \to 0$ (see also [36]). Here $C_J$ is the current central charge of the CFT, which is proportional to the universal DC conductivity $\sigma = \frac{\pi}{16} C_J$ [46]. This is the consequence of conformal symmetry and current conservation. Our previous QMC at O(2) QCP reveals $s/\theta^2 = 0.011(1)$, consistent with the exact value $\frac{C_J}{(4\pi)^2} = 0.01145$ [47–50]. Another feature common to all known examples of disorder operators is that $s$ is always

positive. The positivity of $s$ is proven for the Rényi entropy, i.e. disorder parameter of the replica symmetry in unitary CFTs [51, 52]. In the present case, we generalize the argument in [51] to show that $s(\pi)$ must be positive in a unitary CFT (see Appendix D for details [42]). As we will see below, $s(\theta)$ for DQC follows the same scaling behavior at small $\theta$, but the large $\theta$ behavior is dramatically different. Moreover, we note that the system sizes accesssed here with the disorder operator is much larger than those of the entanglement entropy, simply because the $|\langle X_M \rangle|$ is a equal-time measurement without invoking the replicas.

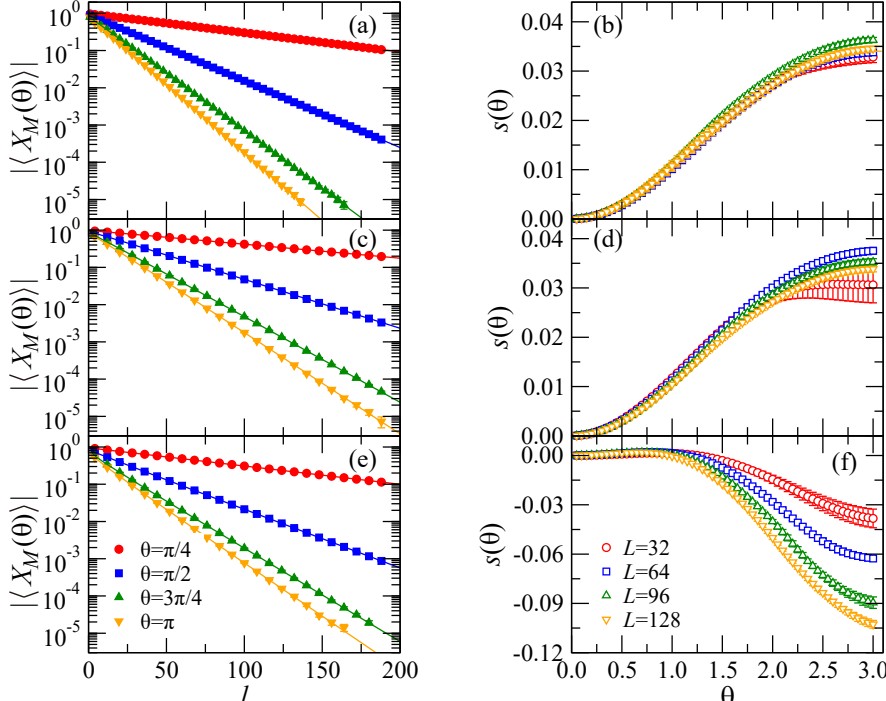

Figure 2: Disorder parameter $|\langle X_M(\theta)\rangle|$ as a function of the perimeter $l = 4R-4$ with system size $L = 96$ at the QCPs for $H_{\text{bilayer}}$ model (a), $H_{J_1-J_2}$ model (c) and the $H_{J-Q_3}$ model (e). (b), (d) and (f) show the obtained $s(\theta)$ with system size $L = 32, 64, 96, 128$ for the three models in (a), (c) and (e), respectively. The convergence of the data with increasing $L$ is clear from the figures.

## 4   Numerical results

We choose the region $M$ to be a $R \times R$ square region in the lattice, with perimeter $l = 4R-4$. Firstly, we compute the disorder parameter $X_M(\theta)$ as a function of perimeter $l$ at the 3D O(3) QCP ($(J_2/J_1)_c = 2.5220$) [38, 39] of the $H_{\text{bilayer}}$ model with system size $L = 32, 64, 96, 128$. Plots of $|\langle X_M(\theta)\rangle|$ v.s. $l$ for representative values of $\theta$'s are shown in Fig. 2(a). Fitting the data with Eq. (4), we obtain the coefficient $s(\theta)$ of the corner correction term, as shown in Fig. 2 (b) [1]. The behavior of $s$ is qualitatively similar to that of the O(2) transition studied

---

[1] we note that although a simple exponential function might also be able to fit the data, but we find in generial for the $\theta$ we have investigated, the goodness of the fit, $\chi^2$, is usually two magnitude larger than those

in [33]. We will mainly use the results from $H_{\text{bilayer}}$ as a reference for the O(3) CFT.

Next, we perform the same QMC simulations for the $H_{J_1-J_2}$ model at its QCP $(J_2/J_1)_c =$ 1.90951(1) [40]. Although the critical theory is the same 3D O(3) CFT, because of the doubling of the unit cell due to alternating $J_1$ and $J_2$ bonds, the disorder parameter $\langle X_M(\theta) \rangle$ exhibits even-odd oscillation as a function of $R$, see Appendix C for details [42]. This is because the boundary of the region $M$ cuts different types of bonds for even and odd $R$: for odd $R$, one of the boundary segments along $y$ always cuts strong $J_2$ bonds, while for even $R$ depending on the exact position of $M$ the boundary may or may not cut strong bonds. Such singlet cutting increases the leading perimeter contribution in the disorder parameter, introducing significant finite-size error when extracting the subleading corner term $s$. For the $J_1$-$J_2$ model, we find that the correct results for $s(\theta)$ (compared to $s(\theta)$ extracted from the bilayer Heisenberg model, free of such complications) can only be obtained from the scaling analysis when disorder operators are constructed on regions whose boundary does not cut any strong bonds. More details of the analysis can be found in Appendix C. We believe this is a general phenomenon and to mitigate finite-size error similar selection of regions must be applied whenever there exists bond order breaking the translation symmetry either explicitly or spontaneously. This is the most important lesson learnt from the study of the $J_1$-$J_2$ model.

Now we turn to the $J$-$Q_3$ model. Because of the VBS order, we would like to design the boundary of $M$ in such a way that it cuts least strong singlet bonds. However, since the VBS order in the $J$-$Q_3$ model forms spontaneously, the pattern of stronger singlet bonds is not known a priori. To overcome this issue, we follow the following procedure: for each measurement, first we calculate the value of the VBS order parameter $(D_x, D_y)$ for the spin configuration and then adjust the region $M$ according to the profile of the instantaneous VBS order to avoid cutting the stronger bonds, as illustrated in Fig. 8. More details can be found in Appendix C [42]. All results below are obtained with this method.

Let us start from small $\theta$. In Fig. 3, we show the fit of the corner correction $s(\theta)$ for small $\theta(\leq 0.25)$ with $s(\theta) = \frac{C_J}{(4\pi)^2}\theta^2$. For the $H_{\text{bilayer}}$ and $H_{J_1-J_2}$ Heisenberg models, we obtain $C_J/(4\pi)^2 = 0.0120(2)$ and $C_J/(4\pi)^2 = 0.0116(2)$, respectively. These values are consistent with $C_J/(4\pi)^2 = 0.01147$ of the O(3) CFT from numerical bootstrap [53] within errorbars. However, as shown in Fig. 3 (e) and (f), the same analysis for the DQC yields a smaller value $C_J/(4\pi)^2 = 0.0088(2)$ A small $C_J$, or equivalently a small DC conductivity $\sigma$, suggests that the theory is more strongly coupled (so the value deviates significantly from that of a free boson).

Most interestingly, we find that the $s(\theta)$ for DQC becomes negative for large $\theta$ as shown in Fig. 2 (f). We also note that $s(\theta)$ become negative for large $\theta$ as the system size increases up to $L = 128$. Such negative values of $s(\theta)$ in DQC are drastically different from the behavior of $s$ observed in all other QCPs investigated so far, including Ising [32], O(2) [33] and also the two different realizations of the O(3) CFT in Fig. 2 (b) and (d). This list can be expanded to include Rényi entanglement entropy as a disorder parameter of the replica symmetry, and it is known that the corner correction $s$ for Rényi entropies must be positive for all unitary CFTs [51, 52]. In fact, we can generalize the argument in [51] to show that $s(\theta = \pi) > 0$ (essentially for any $\mathbb{Z}_2$ symmetry disorder parameter, see Appendix D for details). Therefore a negative $s$ implies strong deviation of the model from unitary CFTs. This is intriguing as measurements of local observables at DQC in the $H_{J-Q_3}$ model appear to exhibit conformal invariance, at least for system sizes accessible to current numerical simulations. Thus our

---

obtained from the fitting form in Eq. (4). We show more details in Appendix A

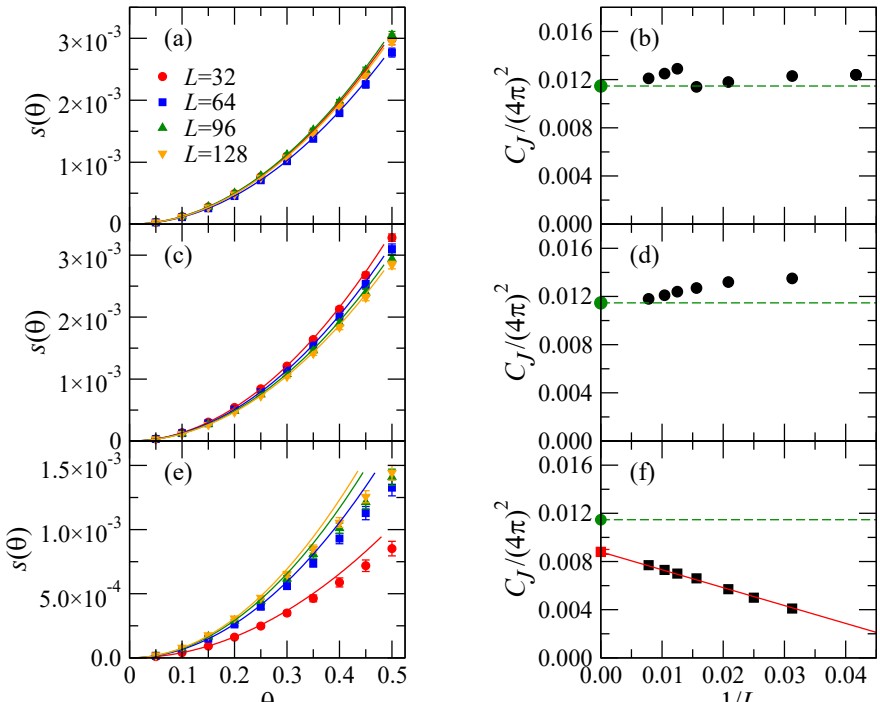

Figure 3: The coefficient of the corner correction $s(\theta)$ for small values of $\theta$ with system size $L = 32, 64, 96, 128$ at the QCPs for $H_{\text{bilayer}}$ model (a), $H_{J_1-J_2}$ model (c) and $H_{J-Q_3}$ model (e). The lines are the data fitting with $s(\theta) = \frac{C_J}{(4\pi)^2}\theta^2$. (b), (d) and (f) show the extrapolation of the obtained $C_J/(4\pi)^2$ as the system size $L$ increases. In case of (b) and (d), the extrapolated $C_J/(4\pi)^2$ approach the theoretical value 0.011 for the O(3) CFT denoted by the green dots and dashe lines. In (f), $C_J/(4\pi)^2$ for DQC apparently extrapolates to a much smaller number (red square and dashed line) compared with the O(3) value (the green dot and dashed line).

observation of a negative $s(\pi)$ provides direct and unambiguous evidence for the breakdown of a unitary CFT description.

## 5 Discussions

Through large-scale QMC simulations and finite-size analyses, we determine the scaling behavior of the disorder operator for $U(1)_{S_z}$ symmetry at the DQCP in the $J$-$Q_3$ model. Most noticeably, the universal corner correction $s$ of the DQC becomes negative, in sharp contradiction to the positivity of $s(\pi)$ in any unitary relativistic conformal field theory. We also observe that the obtained current central charge of DQC is smaller than the typical value of $O(n)$ CFTs.

Our findings, in particularly the negative $s$, raise a number of significant questions about the theory of DQC. One possible explanation for the negative $s$ is that the observed regime of the DQC is actually controlled by a non-unitary CFT, with a (complex) fixed point very close to the physical parameter space. So within a large length scale conformal invariance can still manifest. This possibility has been proposed theoretically in several recent works [6,15–18,20], to explain unusual finite-size scaling behavior from previous numerical simulations [6, 7, 15] and the tension between the numerically observed critical exponents with conformal bootstrap bounds [16]. Our result points to a distinct aspect of this putative non-unitary fixed point, that the universal correction $s$ must be negative.

However, the scenario of complex fixed point implies that the fixed point should be located close to the physical parameter space, in order to explain the large conformal window observed numerically. As a result, it is reasonable to expect that the violation of unitarity in various universal quantities should appear as small complex corrections, which manifest in scaling violation. This is indeed the case in known solvable examples of weakly first-order transition controlled by a complex CFT, such as the $Q = 5$ Potts model in (1+1)d where critical exponents and central charge [54, 55] acquire complex corrections at the actual fixed point. The critical point is the self-dual point of the lattice model, so the disorder operator for the $\mathbb{Z}_Q$ symmetry is related to the $\mathbb{Z}_Q$ Potts spin operators via Kramers-Wannier-type duality. Thus the expectation value of the disorder operator decays as a power law with the length of the interval on which the disorder operator is defined, analogous to the logarithmic corner correction in (2+1)d. For $Q = 5$, the scaling dimension of the spin operator becomes complex: $\Delta_\sigma \approx 0.067 \pm 0.01i$ [55]. If one measures the disorder operator in the $Q = 5$ model, we expect that within the conformal window, the decay is still mainly controlled by the real part of $\Delta_\sigma$, but with small drifts of exponents . Therefore, within the conformal window the result is well-approximated by a power law with positive exponent. Generalizing to (2+1)d, one would expect that $s$ measured from numerical simulations inside the conformal window should still be positive at a weakly first-order transition controlled by a nearby complex CFT, which is not what we have seen.

In light of the situation, it is important to gain more systematic understanding of how the complex fixed point affects the disorder operator and the corner correction $s$, especially their scaling behavior. It is also worthwhile to consider alternative scenarios other than that of a complex fixed point. One proposal is that a dangerously irrelevant operator changes the scaling behavior [7, 15]. How the behavior of the disorder operator is affected remains to be studied. More recently, there emerges new evidence that shows the DQC is a multicritical

point [19]. More thorough investigations of scaling in such modified models are needed to verify the theoretical [23] and numerical predictions [19] and to find better scenarios for the strongly violaition of unitarity we have observed. It will be interesting for future studies to explore other non-local observables, such as Rényi entropies, and consider other microscopic realizations of DQC where dangerously irrelevant operators are absent [24, 56]. From here, more comprehensive studies of DQC are called for.

## Acknowledgement

We are grateful to Hui Shao for sharing unpublished data on the $J$-$Q_3$ model. We would like to thank Cenke Xu, Chao-Ming Jian and Fakher Assaad for comments on a draft. M.C. thanks Yi-Zhuang You for enlightening conversations on DQC. Y.C.W. acknowledges the supports from the NSFC under Grant No. 11804383 and No. 11975024, the NSF of Jiangsu Province under Grant No. BK20180637, and the Fundamental Research Funds for the Central Universities under Grant No. 2018QNA39. N.M. acknowledges the supports from the NSFC under Grant No. 12004020. M.C. acknowledges support from NSF under award number DMR-1846109 and the Alfred P. Sloan foundation. Z.Y.M. acknowledges support from the Research Grants Council of Hong Kong SAR of China (Grant Nos. 17303019, 17301420, 17301721 and AoE/P-701/20), the K. C. Wong Education Foundation (Grant No. GJTD-2020-01) and the Seed Funding "Quantum-Inspired explainable-AI" at the HKU-TCL Joint Research Centre for Artificial Intelligence. We thank the Computational Initiative at the Faculty of Science and the Information Technology Services at the University of Hong Kong and the Tianhe platforms at the National Supercomputer Center in Guangzhou for their technical support and generous allocation of CPU time.

## Appendix A  Fitting analysis

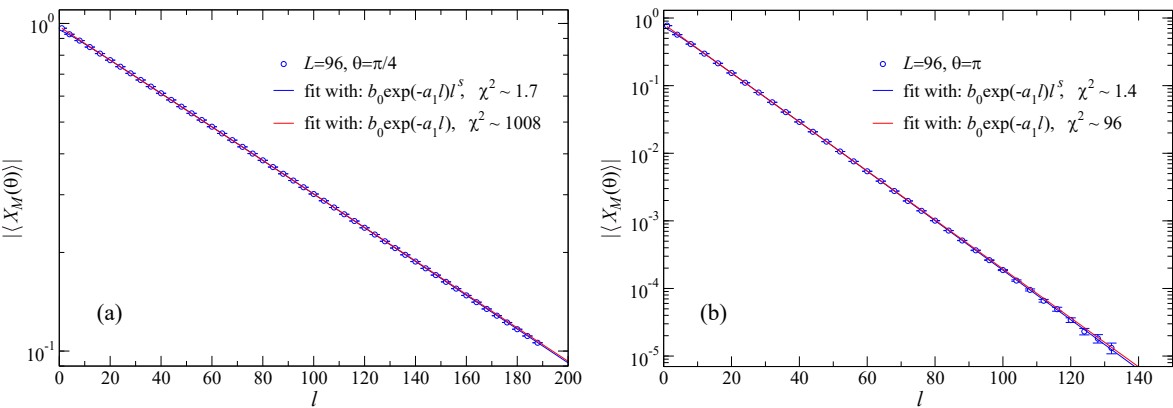

Figure 4: Disorder parameter $|\langle X_M(\theta)\rangle|$ as a function of the perimeter $l = 4R - 4$ with system size $L = 96$ at the QCPs for $H_{\text{bilayer}}$ model at (a) $\theta = \pi/4$ and (b) $\theta = \pi$, resectively. The difference in the quality of the fit in terms of the $\chi^2$, reveal that Eq. (5) is the better choice.

In this appendix, we perform a fitting analysis and compare a purely exponential fit and a fit with a power-law $l^s$ correction:

$$|X_M| \sim b_0 \exp{(-a_1 l)} l^s, \tag{5}$$

or

$$\ln|X_M| \sim a_0 - a_1 l + s \ln(l). \tag{6}$$

As shown in Fig. 4 below, we fit the data (from $l_{min} = 12$) with two different functions: the first one is Eq. (5), i.e., $|\langle X_M(\theta)| = b_0 \exp{(-a_1 l)} l^s$, which gives $b_0 = 0.9597(5)$, $a_1 = 0.01191(5)$, $s = 0.0073(3)$, with $\chi^2 = 1.7$ for $\theta = \pi/4$ and $b_0 = 0.750(1)$, $a_1 = 0.0847(1)$, $s = 0.0364(8)$, with $\chi^2 \sim 1.4$ for $\theta = \pi$; the second one is a purely exponential fit, i.e., $|\langle X_M(\theta)| = b_0 \exp{(-a_1 l)}$, which gives $b_0 = 0.9779(5)$, $a_1 = 0.01176(4)$, with $\chi^2 \sim 1008$ for $\theta = \pi/4$ and $b_0 = 0.815(2)$, $a_1 = 0.0834(2)$, with $\chi^2 \sim 96$ for $\theta = \pi$. Although both functions can go through the data points, the different fitting quality indicated by at least two order of magnitudes difference in the $\chi^2$ clearly shows, that our choice of the form in Eq. (5) is the right fitting form.

## Appendix B  Determination of the deconfined quantum critical point

In this appendix we determine the location of the DQC of the $J$-$Q_3$ model using finite-size scaling. From the scaling hypothesis we know that any dimensionless quantity $O$ measured in a finite-size system fulfills

$$O(q, L) = g[(q - q_c)L^{1/\nu}, L^{-\omega}] \tag{7}$$

with $q_c = [Q/(J + Q)]_c$ the phase transition point in the thermodynamic limit, $\nu$ the critical exponent of correlation length and $\omega$ the correction exponent which generally defers in different microscopic models. It is obvious that if $\omega = 0$ the dimensionless quantity $O$ obtained from systems of different sizes are the same at $q_c$, therefore all curves of $O(q, L)$ as functions of $q$ for different sizes cross at one point, which is the critical point. However, the correction term is not always absent and in general the curves do not actually cross at one point. Thus, we can make use of all the crossings obtained from different curves and find $q_c$ at thermodynamic limit with the following relation,

$$q^*(L) = q_c(\infty) + aL^{-\omega - 1/\nu} \tag{8}$$

where $q^*(L)$ stands for the crossing point of two curves from size $L$ and $L'$. In our study of the $J$-$Q_3$ model $q = Q/(Q + J)$ and we measure spin stiffness $\rho_s$ and Binder ratios of the order parameters for Néel ($R_s$) and VBS ($R_d$) orders. The spin stiffness $\rho_s$ is calculated from the winding number,

$$\rho_s = \frac{1}{2\beta}(W_x^2 + W_y^2) \tag{9}$$

which has the following scaling form:

$$\rho_s = L^{-z} f(qL^{1/\nu}). \tag{10}$$

In $J$-$Q_3$ model the dynamical exponent $z = 1$ and $\rho_s L$ is therefore dimensionless. As for the Binder ratio,

$$R_s = \frac{\langle m_{sz}^4 \rangle}{\langle m_{sz}^2 \rangle^2}, \qquad R_d = \frac{\langle D^4 \rangle}{\langle D^2 \rangle^2} \qquad (11)$$

where $m_{sz} = \frac{1}{L^2}\sum_{x,y}(-1)^{x+y}S_{x,y}^z$ is the staggered magnetization $m_s$ along the $z$ (quantization) axis, and $D^2 = D_x^2 + D_y^2$, is the VBS order parameter with $D_x = \frac{1}{L^2}\sum_{x,y}(-1)^x\mathbf{S}_{x,y}\cdot\mathbf{S}_{x+1,y}$ and $D_y$ defined analogously [57]. Both of those two ratios are dimensionless. We perform QMC simulations on the $J$-$Q_3$ model and determine the $q$ dependence of the dimensionless quantities $\rho_s L$, $R_s$ and $R_d$ as shown in Fig. 5(a), (b) and (c). After that, we calculate all the $q^*(L)$ as the crossing point of $(L, 2L)$ from three different quantities using system sizes from $L = 8$ to $L = 128$. All the crossings are depicted in Fig. 5(d), fitted by Eq. (8). From the fitting results in these three different observables we can get the $q_c(\infty) = 0.59864(5)$ which is the value used in the main text.

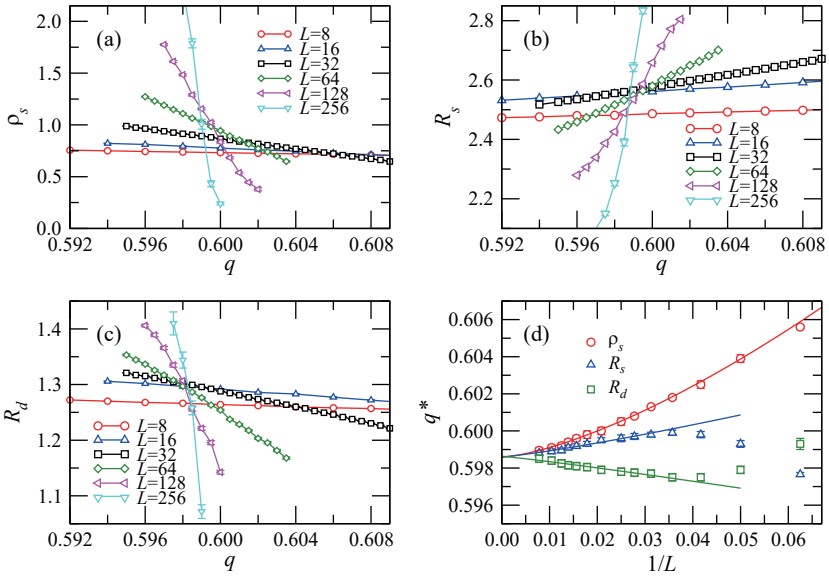

Figure 5: The $q = [Q/(J + Q)]$ dependence of JQ$_3$ model for $\rho_s L$ (a), $R_s$ (b) and $R_d$ (c) with $L = 8, 16, 32, 64, 128$ and $256$. All the crossings of two sizes $L$ and $2L$ for $\rho_s L$, $R_s$ and $R_d$ are presented in (d). The curves are fitting function in Eq. (8) with $q_c = 0.59862(5)$ for $\rho_s L$, $q_c = 0.59863(5)$ for $R_s$, and $q_c = 0.59865(5)$ for $R_d$.

## Appendix C   The choice of region $M$

In this appendix, we discuss how to choose the region $M$ to obtain the correct scaling behavior of $\langle X_M \rangle$. As shown in the main text, for $H_{\text{bilayer}}$, since there is no translation symmetry breaking (explicit or spontaneous), the choice of the region is immaterial. However, the $J_1 - J_2$ case is different because the alternating strenghs of Heisenberg couplings doubles the unit cell, as shown in Fig. 6. If the region $M$ is a $R \times R$ square (green shaded region in Fig. 6), $M$ with even or odd $R$ cut very different types of bonds on the boundary.

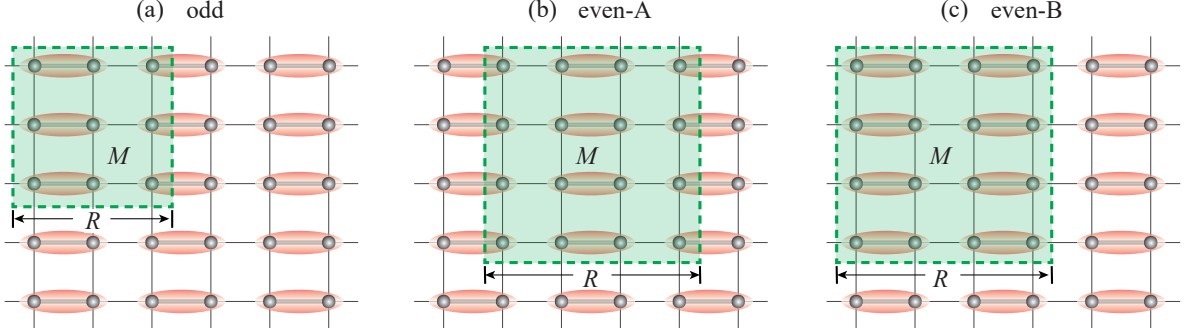

Figure 6: Three types of region $M$: (a) region $M$ with odd $R(=3)$, whose boundary cuts one column of strong singlet bonds; (b) region $M$ with even $R(=4)$ and cutting two columns of strong singlet bonds; (c) region $M$ with even $R(=4)$ and cutting no strong singlet bonds.

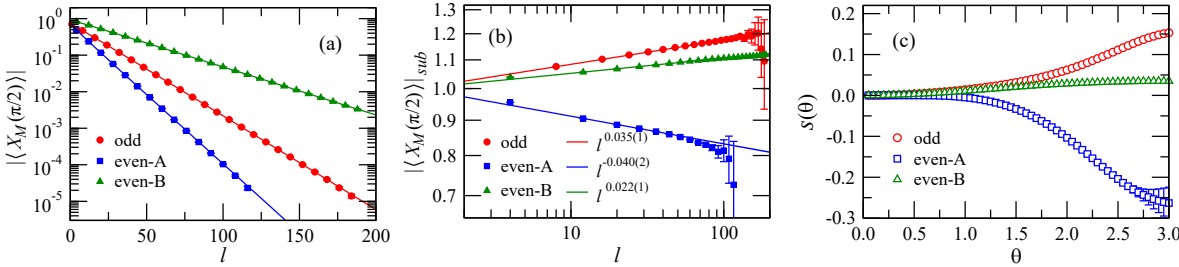

Figure 7: $|\langle X_M(\theta)\rangle|$ for the $H_{J_1-J_2}$ model, all obtained with $L=96$: (a) the disorder parameter and (b) its subleading term $|\langle X_M(\theta)\rangle|_{sub}$ as functions of $l$ at $\theta = \pi/2$, for the three types of the region $M$. (c) $s(\theta)$ for the three types of the region $M$, respectively.

More concretely, when $R$ is odd (e.g. $R = 3$ in Fig. 6 (a)), the boundary of $M$ inevitably cuts one column of $J_2$ bonds. When $R$ is even, as shown in Fig. 6 (b) and (c), depending on the exact location of $M$, the boundary can cut two columns of $J_2$ bonds (as in (b)) or avoid cutting $J_2$ bonds at all (as in (c)). Numerically, we find that the three choices of $M$ yield distinct values and scaling behavior of the disorder parameter $\langle X_M \rangle$. Only for those regions cutting no $J_2$ bonds, finite-size scaling analysis converges to the expected result for the $(2+1)d$ O(3) CFT. The issue can be clearly seen from the representative data in Fig. 7. Fig. 7 (a) shows the $|\langle X_M(\theta) \rangle|$ at $\theta = \pi/2$ for $L = 96$ at the QCP of $H_{J_1-J_2}$. Disorder parameters corresponding to three different boundaries as illustrated in Fig. 6 (a), (b) and (c), denoted as odd, even-A and even-B in the figure all show different perimeter dependence. Even-A type boundaries show the largest linear coefficient in the perimeter contribution, which makes sense as this type of boundary cuts the most $J_2$ bonds. While the perimeter law is non-universal, the dependence on the details of the boundary also manifests in the corner correction, at least in our finite-size analysis. In order to show obviously the subleading term from corner correction, we also extract the subleading term of the disorder parameter $|X_M|_{sub} = |X_M|/[b_0 \exp{(-a_1 l)}] = l^s$, as shown in Fig. 7 (b). The problem is clearly illustrated in Fig. 7 (c), where $s(\theta)$ extracted from the disorder parameters computed using the three types of boundaries are shown for system size $L = 96$. Here one sees that $s$ for even-A boundary becomes negative, which violates the positivity constraint at $\theta = \pi$ and is obviously unphysical. We believe that the relatively large perimeter contribution for even-A boundary data strongly affects the precision of the fitting, since the corner contribution is subleading to the perimeter term. For the other two types of boundaries (odd and even-B), now both give positive $s(\theta)$, but the $\theta$ dependence is still quite different. We find it is the even-B type regions that give the correct value of the current central charge of the O(3) CFT, from fitting $s(\theta) \approx \frac{C_J}{(4\pi)^2}\theta^2$. This analysis suggests that to mitigate the finite-size error in the data fitting, one should construct disorder operators on regions which minimize the perimeter contribution. In particular, when there is lattice symmetry breaking induced by bond or plaquette order, the strong bonds should be avoided.

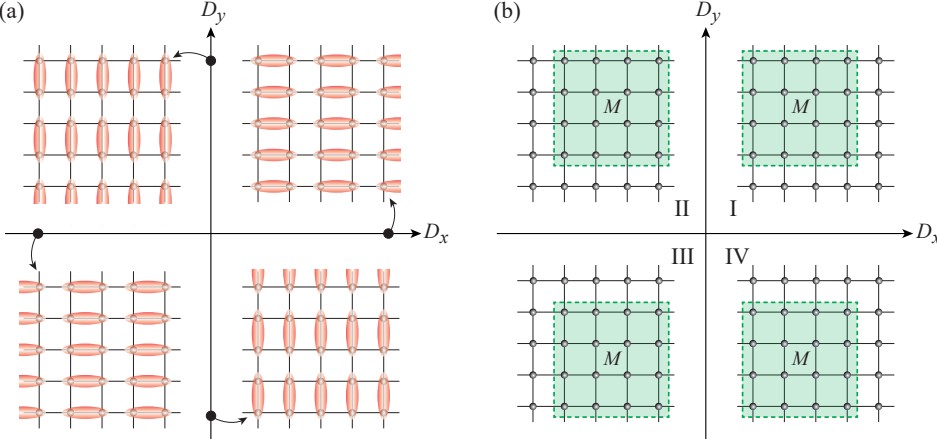

Figure 8: The choice of region $M$ with " even-$\widetilde{B}$ " type bounary: (a) The four special patterns of VBS order in the plane of order parameter $(D_x, D_y)$. (b) Porper region $M$ with even $R(=4)$ in each quadrant, whose boundary cuts least column of singlet bonds.

In case of the DQC, although $H_{J-Q_3}$ is translation-invariant, it is already well-known that finite-size analysis of correlations can be tricky due to the domains of VBS formed

spontanenously when $Q \geq Q_c$. For the disorder operator, boundary dependence similar to those observed in the $J_1 - J_2$ model also shows up in the naive measurements of $\langle X_M \rangle$ at DQC. To minimize the effect of cutting strong bonds due to residual VBS order, we take a choice of region $M$ with " even-$\widetilde{B}$ " type bounary: firstly, we calculate the two components of the VBS order $(D_x, D_y)$ with $D_x = \frac{1}{L^2} \sum_{x,y} (-1)^x \mathbf{S}_{x,y} \cdot \mathbf{S}_{x+1,y}$ and $D_y$ defined analogously for each given spin configuration (one microstate or one sample in the SSE QMC) at the $S_z$ basis, and then adjust the region $M$ according to the profile of the instantaneous VBS order, as illustrated in Fig. 8. For example, if $D_x > 0$, and $D_y > 0$, i.e., the first quadrant, we will choose the region $M$ as shown in Fig. 8(b)-I. The similar choice works for the case of the other three quadrants. With such a setup, we then measure the $\langle X_M \rangle$ with " even-$\widetilde{B}$ " type boundary. A comparison of measurements with three different boundaries is given in Fig. 9. We find that this method achieves the most robust convergence of $\langle X_M \rangle$ at DQC, as shown in the main text.

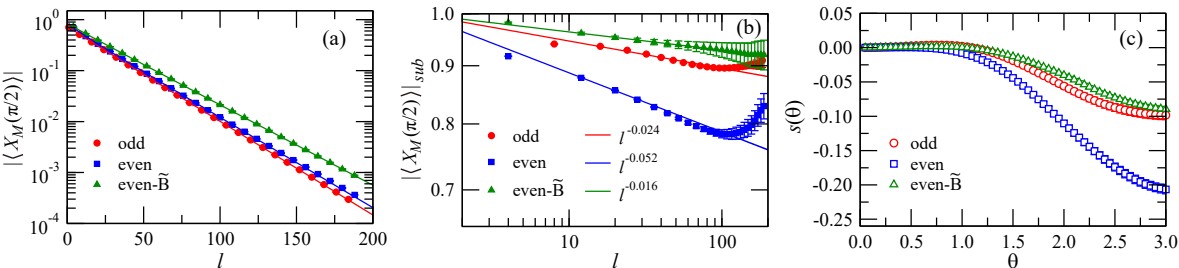

Figure 9: $|\langle X_M(\theta) \rangle|$ for the $H_{J-Q_3}$ model at DQC, all with $L = 96$: (a) the disorder parameter and (b) its subleading term $|\langle X_M(\theta) \rangle|_{sub}$ as functions of $l$ for $\theta = \pi/2$ for the three types of the region $M$, respectively. (c) $s(\theta)$ for the three types of the region $M$, respectively. It is interesting to note here that all three different types of $M$ lead to negative $s(\theta)$ when $\theta$ is close to $\pi$.

In order to show how our results are sensitive to the parameter $q = Q/(Q + J)$, we also calculate the disorder operator for different $q$ in the vicinity of DQC and get the coefficient of the logarithmic corner correction, as shown in Fig. 10. We can find different scaling behavior in both symmetry-breaking phases and at the DQC manifest, and therefore, our results at $q_c = 0.59864$ indeed reflect the critical properties of the system.

## Appendix D   Positivity constraint on $\mathbb{Z}_2$ disorder parameter

In [51] it was shown that in a general unitary quantum field theory (QFT), Rényi entropies satisfy the following inequality:

$$\det\left(\left\{ e^{-(n-1)S_n(M_i \cup \bar{M}_j)} \right\}_{i,j=1,\ldots,m}\right) \geq 0. \tag{12}$$

Here $M_i, i = 1, \ldots, m$ is a collection of (codimension-1) regions in the half space of positive Euclidean time, and $\bar{M}_j$ is the Euclidean time-reflected regions corresponding to $M_j$.

We now prove a similar inequality for $\mathbb{Z}_2$ disorder operator. Suppose that $U_g$ is a $\mathbb{Z}_2$ symmetry in the QFT (i.e. $U_g^2 = 1$ and therefore $U_g$ is hermitian), which is represented by a

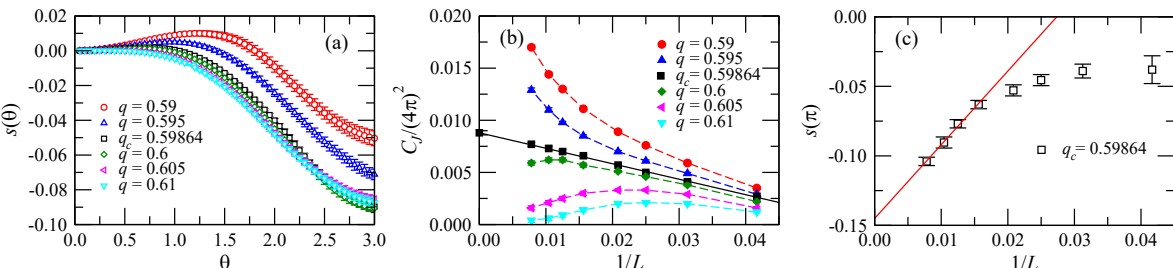

Figure 10: (a) The $s(\theta)$ as functions of $\theta$ with system size $L = 96$ and (b) the obtained $C_J/(4\pi)^2$ as a function of $1/L$ for different $q$ in the vicinity of DQC. The different scaling behaviors in the two symmetry-breaking phases and at the DQC manifest. (c) The $s(\pi)$ as functions of $1/L$ for $q_c = 0.59864$.

topological surface operator in the Euclidean spacetime. The disorder parameter for a region $M$ is then given by the (suitably normalized) path integral with an insertion of an open surface operator $X_M$, which is just $U_g$ restricted on $M$. Following [51], we split the path integral for positive and negative Euclidean time. Consider a family of $M_i$ in the positive Euclidean time half-space, and write $\phi^{\pm}$ for fields restricted to positive and negative Euclidean time. We have

$$
\sum_{i,j}\lambda_i\lambda_j^*\langle X_{M_i\cup\bar{M}_j}\rangle = \mathcal{N}^{-1}\sum_{i,j}\lambda_i\lambda_j^*\int^{M_i\cup\bar{M}_j}\mathcal{D}\phi\,e^{-\mathcal{S}[\phi]}
$$

$$
= \mathcal{N}^{-1}\int\mathcal{D}\phi_0(\mathbf{x})\left(\sum_i\lambda_i\int_{\phi^+(0,\mathbf{x})=\phi_0(\mathbf{x})}^{M_i}\mathcal{D}\phi^+\,e^{-\mathcal{S}[\phi^+]}\right)\left(\sum_j\lambda_j^*\int_{\phi^-(0,\mathbf{x})=\phi_0(\mathbf{x})}^{\bar{M}_j}\mathcal{D}\phi^-\,e^{-\mathcal{S}[\phi^-]}\right).
$$
(13)

Here the subscript $M$ indicates insertions of the corresponding open surface operators in the path integral. Equivalently, one may view the insertion as changing the boundary condition of the fields along the surface $M$. The normalization factor $\mathcal{N} = \int\mathcal{D}\phi\,e^{-\mathcal{S}[\phi]}$ is just the path integral without any open surface inserted. $\phi_0$ is the common value of $\phi^+$ and $\phi^-$ at Euclidean time $\tau = 0$. If the action has the time-reflection symmetry $\mathcal{S}[\phi(\tau,\mathbf{x})] = \mathcal{S}[\phi(-\tau,\mathbf{x})]^*$, together with the hermiticity of the inserted operator, a change of variables $\phi(\tau,\mathbf{x}) \to \phi(-\tau,\mathbf{x})$ in the path integral proves that the two terms in the brackets are complex conjugate of each other, and the result is positive. Since $\lambda_i$'s are arbitrary complex numbers, the condition is equivalent to

$$
\det\left(\left\{\langle X_{M_i\cup\bar{M}_j}\rangle\right\}_{i,j=1,\ldots,m}\right) \geq 0.
$$
(14)

In the following we will write $\langle X_M\rangle = e^{-S(M)}$ ($S$ not to be confused with the entropy or the action $\mathcal{S}$ in the path integral). In a CFT in (2+1)d, $S(M)$ should take the following form:

$$
S(M) = a_1|\partial M| - s\ln\frac{l_M}{\delta} + a_0 + O\left(\frac{\delta}{l_M}\right).
$$
(15)

Here $\partial M$ denotes the boundary of $M$, and $|\partial M|$ is the perimeter of the region. $l_M$ is the linear size of $M$ and $\delta$ is a short-distance cut-off. $s$ is the sum of universal constants for each sharp corner of the region.

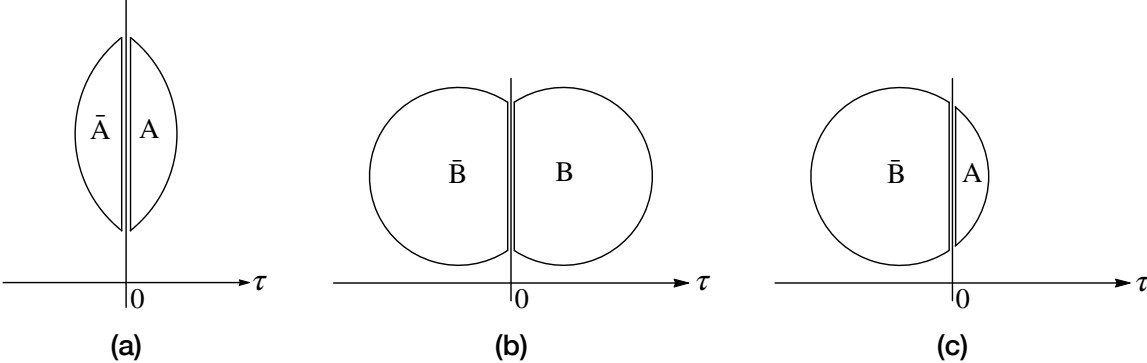

Figure 11: (a) Regions $A$ and $\bar{A}$. $\bar{A}$ is the Euclidean time-reflected image of $A$ with respect to $\tau = 0$. The two regions touch each other at the $\tau = 0$ plane, and we take the limit of no separation. Together $A \cup \bar{A}$ has two sharp corners. (b) Regions $B$ and $\bar{B}$. (c) Regions $A$ and $\bar{B}$. $A \cup \bar{B}$ has no corners.

When $m = 2$, denote the two regions by $A$ and $B$, the inequality reduces to

$$2S(A \cup \bar{B}) \geq S(A \cup \bar{A}) + S(B \cup \bar{B}). \tag{16}$$

Now we choose the two regions $A$ and $B$ as shown in Fig. 11. The region $A \cup \bar{A}$ has two sharp corners with the same opening angles $\alpha$, while $B \cup \bar{B}$ has two corners with opening angles $2\pi - \alpha$. It can be easily checked that

$$|\partial(A \cup \bar{A}| + |\partial(B \cup \bar{B})| = 2|\partial(A \cup \bar{B})|, \tag{17}$$

where $|\partial(V)|$ is the perimeter of the region $V$. Thus the perimeter terms in $S$ all cancel out. We then note that $A \cup \bar{B}$ has a smooth boundary with no corners. Notice that $A$ and $B$ can be considered to have the same linear size $l$. So in order for the inequality to hold for arbitrary linear size of the region, generally we must have

$$-2s(\alpha) \ln \frac{l}{\delta} - 2s(2\pi - \alpha) \ln \frac{l}{\delta'} + \text{const.} \leq 0. \tag{18}$$

It is not difficult to show that $s(\alpha) = s(2\pi - \alpha)$, therefore in order to satisfy Eq. (18) for arbitrarily large $l$, $s(\alpha)$ must be positive.

A slight generalization of the argument, with $B$ having opening angle $\beta$ instead of $2\pi - \alpha$, gives

$$s(\alpha) + s(\beta) \geq 2s\left(\frac{\alpha + \beta}{2}\right). \tag{19}$$

Together with $s(\alpha = \pi) = 0$ we can see that $s(\alpha)$ is a non-negative, decreasing and convex function of $\alpha$ for $0 \leq \alpha \leq \pi$. Similar conclusions for Rényi entropies were obtained in [52].

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
