# Peer review of "Scaling of disorder operator at deconfined quantum criticality"

_SciPost Physics_

## Round 2 · Referee Report · Anonymous (Referee 1) · 2022-3-1

Strengths

  1. Timely study of a new aspect of the DQC scenario.
  2. Marked differences in the behavior of the disorder operator mean value scaling between conventional QCP and anticipate DQC point are clearly exposed.
  3. A careful numerical analysis is provided.
  4. Details of the measurement scheme in the presence of translational symmetry breaking are provided.

Weaknesses

  1. A few details need to be better explained, as detailed below.
  2. The paper formatting needs to be adapted.
  3. Several language flaws need to be corrected (see list below).

Report

The authors present a detailed numerical study using QMC methods of the scaling behavior of appropriate disorder operators in several quantum spin models at a quantum phase transition out of an AF ordered phase. In all cases the authors fit their numerical data, obtained on varying finite-size systems and square subregions, to a scaling form that contains a logarithmic corner contribution. They then carefully analyze the dependence of the corner contribution on the U(1) rotation angle entering the disorder parameter. For the bilayer and columnar dimer model (referred to as the J1-J2 model in the manuscript) the observed behavior is in accord with earlier results by the same authors on related models and the low-angle scaling is in agreement with universal predictions from conformal symmetry and current conservation.

Interestingly, for the anticipated DQC in the J-Q3 model, the behavior differs in two important aspects: (i) the extrapolated current central charge falls significantly below the free boson value, indicative of strong coupling, (ii) the large-angle value becomes negative, and the authors argue that this is not possible in any unitary CFT. This finding in certainly very interesting and adds an important new piece of information to the DQC scenario, in particular in view of several recent developments, as detailed by the authors.

In addition to the physical results, the authors provide detailed explanations of how to measure the disorder parameter mean value in the presence of (explicit or spontaneous) translational symmetry breaking, and such technical details are certainly very useful.

I find that all the general criteria for publication in SciPost Phys. are already met by the manuscript or will be met after my requests for changes are implemented, and also the expectation No. 3 is fulfilled, since this work opens a new perspective and approach to study the DQC scenario, calling for follow-up work from both computational and field-theoretical perspectives, e.g., on related models and transitions.

In summary, I suggest to publish this work after the following changes and clarifications are performed on the current manuscript:

Requested changes

  1. All the SM should be turned into regular appendices or integrated into the main text.

  2. The authors should provide details regarding the quality of the fits of their data to Eq. (4) in the form of chi^2 values that add information beyond the statement in [49].

  3. The definition of the perimeter l=4R-R should be justified (why not l=4R)?

  4. The authors should explain how they arrived at the estimated scaling dimension for the spin operator in [53]. Furthermore, it should be made more explicit, how the last sentence in [53] relates to that result, i.e., the smallness of the imaginary part (I suppose).

  5. In the current SM subsection on the choice of M, the authors claim that for the Even-A subregion the negative corner contribution comes about because of a large perimeter contribution that affects the fitting prediction. If this was indeed the case, then why does a similar problem not affect/explain the observed negative value of s in the case of the J-Q3 model? Would the large-angel values of s turn out to become positive for the J-Q3 model if better data would be available?

  6. The authors should show ( e.g. included in the current Fig. S5) the results of a naive measurement of <X_M> at the anticipated DQC point in order for the reader to compare to the results from their correction procedure.

  7. Overall, the manuscript contains a large number of language flaws, which must be corrected. Below is an incomplete list focusing on the first page only:

-title: “Scaling of disorder operator…“ -> “Scaling of the disorder operator…“ -page 1: “exciciting“ -> “exciting“ -page 1: “which is defined“ -> “which are defined“ -page 1: “U(1) disorder operator“ -> “The U(1) …“ -page 1: “one side of DQC exhibits the Valence Bond Solid“ -> “one site of the DQC exhibits valence bond solid“ -page 1: “behavior of disorder operator“ -> “behavior of the disorder operator“ -page 1: “on square lattice, using the ubiased Stochastic…“ -> “on the square lattice, using unbiased stochastic …“ …

  • validity: top
  • significance: high
  • originality: high
  • clarity: high
  • formatting: good
  • grammar: good

Author:  Meng Cheng  on 2022-06-20  [id 2598]

(in reply to Report 1 on 2022-03-01)
Category:
answer to question

We thank referee for his/her concise summary and high assessments of the importance of our work, and the useful comments and suggestions. We are more than happy to implement the requested changes and have substantially revised the main text and SM accordingly. Below are our responses to the requests:

Requested changes 1: All the SM should be turned into regular appendices or integrated into the main text.

Reply 1: Thanks for the suggestion and we have turned the SM into regular appendices.

Requested changes 2: The authors should provide details regarding the quality of the fits of their data to Eq. (4) in the form of $\chi^2$ values that add information beyond the statement in [49].

Reply 2: Thanks for the suggestion and we have added a more careful analysis of the fits in the Fig. 4 in the revised Appendix A.

Requested changes 3: The definition of the perimeter $l=4R-R$ should be justified (why not $l=4R$)?

Reply 3: Sorry for the confusion. Since $l$ is the perimeter of region $M$ and the $M$ is a square shaped area with linear length $R$, we use $l=4R-4$ to substract the overcounting of the 4 corners of the $M$. Of course, such a substraction will not affect the scaling behavior in the thermodynamic limit.

Comment 4: The authors should explain how they arrived at the estimated scaling dimension for the spin operator in [53]. Furthermore, it should be made more explicit, how the last sentence in [53] relates to that result, i.e., the smallness of the imaginary part (I suppose).

Reply 4: The scaling dimension is extracted from Ref. [52] ([55] in the revised manuscript). We have also expanded the discussions about the disorder operator in 1+1d Potts model.

Comment 5: In the current SM subsection on the choice of M, the authors claim that for the Even-A subregion the negative corner contribution comes about because of a large perimeter contribution that affects the fitting prediction. If this was indeed the case, then why does a similar problem not affect/explain the observed negative value of s in the case of the J-Q3 model? Would the large-angel values of s turn out to become positive for the J-Q3 model if better data would be available?

Reply 5: Thanks for the question. Actually, as shown in Fig.9 of the revised manuscript (in the appendix), the Even-A indeed affect the scaling behavior, note that even in Fig.9 is the Even-A in the Fig.7 for the J1-J2 model. That is why, we have to introduce the even-$\tilde{B}$ approach for the J-Q3. And as for the large angle value of s, as shown in the Fig.9 (c), both even (the Even-A) and even-$\tilde{B}$ give negative values, in fact the even (the Even-A) is more negative than even-$\tilde{B}$. So we do not think s will turn to positive for J-Q3 at large angle.

Comment 6: The authors should show ( e.g. included in the current Fig. S5) the results of a naive measurement of $<X_M>$ at the anticipated DQC point in order for the reader to compare to the results from their correction procedure.

Reply 6: Thanks for the suggestion. In the revised Fig.10 (in the appendix), we have now included the $s(\pi)$ as a function of $1/L$ for $q_c=0.59864$ at the DQCP.

Attachment:

reply-1.pdf

---

## Round 2 · Referee Report · Anonymous (Referee 2) · 2022-3-2

Report

This paper applies the technology of symmetry disorder operators to deconfined quantum criticality. The paper is extremely interesting and worthwhile and apart from the specific points below it is clear. In addition to obtaining some scaling functions for the O(3) universality class, one main claim is that the line operator shows a violation of conformal invariance for deconfined quantum critical point (DCP) in negativity of quantity s(pi), a striking result.

However, I have a concern about the procedure for defining the line operators in a way that avoids strong bonds and the extent to which finite size effects are controlled. I hope that the authors can clarify these issues to solidify what will then be a nice paper.

  1. Various definitions of the line operator are compared and Figure S3 (supplemental material) shows that for some of these choices of definition the expected behavior is not found. The authors' conclusion is that these choices of definition suffer from severe finite size effects.

What I do not see presented is evidence that similar finite size effects are absent for the preferred definition, in the DCP case where there is no independent result to check against. This is important because the authors explain in S3 that the finite size effects can be of the same magnitude as the phenomenon of interest and can artificially give a negative s(pi).

Therefore, the severity of finite size effects and the error bars in the large L extrapolation should be quantified directly from the data. This might allow the authors to directly support the claim that their chosen definitions are free of the severe finite size effects seen in Figure S3. Alternatively, it is possible that the size of differences between curves in S5 is reflective of the error bar in the final result.

The authors suggest in the supplemental material that the problem with finite size effects could be related to having to fit a nonuniversal leading term and also a universal subleading one. If it is possible with available data, an alternative may be to subtract off the leading term by taking appropriate differences of f(l)=ln |<X>| for different shapes/sizes instead of simply fitting it. For example f(l)-b f(l/b) for some b.

It would also be useful to show in the supplemental material some intermediate plots that give a sense of the error bars involved in extracting the logarithmic coefficient.

  1. The procedure of adjusting the path of the defect operator in response to the VBS configuration is justified on the grounds that it reduces the contribution from strong bonds. If I understand correctly, the adjustment is a simple translation of the operator.

It would seem that this procedure cannot change the coefficient of the leading perimeter law term. This term is set by microscopic correlations. Since the critical VBS order parameter is small at large L, a translation only changes microscopic conditions, on average, by an amount that is small at large L. Therefore, the change in the definition will only affect subleading terms.

For this reason, the argument for this protocol based on the J1-J2 model example does not seem fully conclusive on its own to me. In the J1-J2 model the differences in the protocols affect the leading term. The meaning of a strong bond is also different in the explicitly dimerized model and DCP model.

If point 1 was addressed with evidence quantifying finite size and fitting error then this could avoid needing to rely on this argument.

Other comments:

“is arguably the enigma of…”. I did not understand the exact meaning: this may be my ignorance of a use of the word, otherwise this could be clarified.

Why is the J-Q3 model chosen (rather than J-Q)?

“Previous studies of the (2+1)d Ising and … suggest”: is it a conjecture or has it been derived using RG?

“measurements of local observables in the H_J-Q3 model appear to exhibit conformal invariance.” Is this correct? For a different DCP model in ref 14 the anomalous dimensions from the correlators become negative, violating unitarity bound, at scales roughly comparable with these.

What does it mean to calculate the VBS order parameter for a “spin configuration”?

Though there is an emergent U(1) for the VBS there is no corresponding microscopic onsite symmetry, so there is an obstacle to defining a similar disorder operator. Is there any way around this?

  • validity: -
  • significance: -
  • originality: -
  • clarity: -
  • formatting: -
  • grammar: -

Author:  Meng Cheng  on 2022-06-20  [id 2599]

(in reply to Report 2 on 2022-03-02)
Category:
answer to question

We would like to thank the referee for his/her high assessments, the careful reading and consise summary of both the physical impact and technical (numerical) innovation of our work. Regarding the comments, we respond them one by one below and make the corresponding changes in the revised manuscript.

Comment1: Various definitions of the line operator are compared and Figure S3 (supplemental material) shows that for some of these choices of definition the expected behavior is not found. The authors' conclusion is that these choices of definition suffer from severe finite size effects.

What I do not see presented is evidence that similar finite size effects are absent for the preferred definition, in the DCP case where there is no independent result to check against. This is important because the authors explain in S3 that the finite size effects can be of the same magnitude as the phenomenon of interest and can artificially give a negative $s(\pi)$.

Therefore, the severity of finite size effects and the error bars in the large L extrapolation should be quantified directly from the data. This might allow the authors to directly support the claim that their chosen definitions are free of the severe finite size effects seen in Figure S3. Alternatively, it is possible that the size of differences between curves in S5 is reflective of the error bar in the final result.

The authors suggest in the supplemental material that the problem with finite size effects could be related to having to fit a non-universal leading term and also a universal subleading one. If it is possible with available data, an alternative may be to subtract off the leading term by taking appropriate differences of $f(l)=ln |<X>|$ for different shapes/sizes instead of simply fitting it. For example f(l)-b f(l/b) for some b.

It would also be useful to show in the supplemental material some intermediate plots that give a sense of the error bars involved in extracting the logarithmic coefficient.

Reply 1: We thank the referee for the insightful and professional suggestion and we are sorry that there are misunderstanding in our presentation that makes the referee confused about our data analysis. Let's try to explain in the more details here. We have tried three difference geometry of the region $M$ to obtain the $\langle X \rangle$ for the scaling analysis, the odd, even-A and even-B for the J1-J2 model and the odd, even and even-$\tilde{B}$ for J-Q3 model, these are the Fig.7 and Fig.9 in the revised Appendix.

From these analyses, we find that odd and even-A for J1-J2 model and odd and even for J-Q3, give unexpected behavior which are the "finite size effect" the respected referee mentioned. And we believe the reason of such unexpected behavior is due to the fact the the boundaries of these two ways of defining the region $M$ cut too many local singlet bonds within nearest neighbor and generate too much local entanglement. So it is for both cases, the J1-J2 and J-Q3, "similar finite size effects are absent for the preferred definition", and we are sorry if our previous presentanation were not clear about this point.

Having this in mind, we found that the even-B choice for J1-J2 and even-$\tilde{B}$ for J-Q3 give the similar preferred finite size scaling behavior of $\langle X \rangle$ and it is from here we can read the log-correction for 2+1 O(3) consistent with the Bootstrap results and the large-angle $s$ is negative for DQCP.

As for the other suggestion of the respect referee, we have indeed implemented the $f(l)- bf(l/b)$, i.e., the subleading term of the disorder parameter $|X_M|_{sub}=\frac{|X_M|}{b_0 \exp{(-a_1 l)}}$, and it worked very well, as shown in the Fig.7(b) and Fig.9(b) of revised manuscript.

Comment 2: The procedure of adjusting the path of the defect operator in response to the VBS configuration is justified on the grounds that it reduces the contribution from strong bonds. If I understand correctly, the adjustment is a simple translation of the operator.

It would seem that this procedure cannot change the coefficient of the leading perimeter law term. This term is set by microscopic correlations. Since the critical VBS order parameter is small at large L, a translation only changes microscopic conditions, on average, by an amount that is small at large L. Therefore, the change in the definition will only affect subleading terms.

For this reason, the argument for this protocol based on the J1-J2 model example does not seem fully conclusive on its own to me. In the J1-J2 model the differences in the protocols affect the leading term. The meaning of a strong bond is also different in the explicitly dimerized model and DCP model.

If point 1 was addressed with evidence quantifying finite size and fitting error then this could avoid needing to rely on this argument.

Reply 2: We thank again the referee for the insightful comment, and we totally agree with the referee that ideally, the protocol for J-Q3 (the even-$\tilde{B}$) shall have less influence from the strong bonds compared with the even-B protocol of the J1-J2, because, as the referee rightly put, the critical VBS of J-Q3 shall not affect the leading term. However, as the referee is well aware of, it seems that the domain of the VBS at the DQCP can be actually large, at least for the system sizes we have studied. Therefore, in the revised Fig.9 in the appendix, we can see that the difference between the even-$\tilde{B}$ protocol and say, odd protocol still exist even for the leading term. Of couse, the difference between them are indeed much smaller than that for the J1-J2 case. It is exactly because of such subtle competition of the system sizes we could actually simulate and the interesting but complex behavior of the J-Q3 model, that we are forced to follow the protocol after careful examination.

Comment 3: Other comments: ''is arguably the enigma of..''. I did not understand the exact meaning: this may be my ignorance of a use of the word, otherwise this could be clarified.

Reply 3: We thank the referee for the suggestion and have replaced the sentence to "the complex behavior of the DQCP ...".

Comment 4: Why is the J-Q3 model chosen (rather than J-Q)?

Reply 4: There are several kind of J-Q model : J-Q2, J-Q3, et al, which are only distinguished by the difference of Q term. The reason of choosing J-Q3 is because the critical point is larger in the axis of $J/Q$ such that the QMC simulation can be more efficient.

Comment 5: ''Previous studies of the (2+1)d Ising and … suggest'': is it a conjecture or has it been derived using RG?

Reply 5: A systematic RG derivation has not been done yet, but we notice that Eq. 4 contains all terms compatible with scale invariance. This form is also supported by extensive calculations in other critical theories, as cited in the manuscript.

Comment 6: ''measurements of local observables in the $H_{J-Q3}$ model appear to exhibit conformal invariance.'' Is this correct? For a different DCP model in ref 14 the anomalous dimensions from the correlators become negative, violating unitarity bound, at scales roughly comparable with these.

Reply 6: This is correct, at least from available numerical data.

Comment 7: What does it mean to calculate the VBS order parameter for a ''spin configuration''?

Reply 7: Here, we mean that for each given``spin configuration'' (one microstate or one sample in the SSE QMC) at the $S_z$ basis, we calculate the two components of the VBS order $(D_x, D_y)$ with $D_x=\frac{1}{L^2}\sum_{x,y}(-1)^{x} {\bf S}_{x,y} \cdot {\bf S}_{x+1,y}$ and $D_y$ defined analogously. It could take some confusion, we clarify the description in our revised manuscript.

Comment 8: - Though there is an emergent U(1) for the VBS there is no corresponding microscopic onsite symmetry, so there is an obstacle to defining a similar disorder operator. Is there any way around this?

Reply 8: While there is no corresponding U(1) symmetry, there is a $C_4$ lattice rotation symmetry that gets enhanced to U(1) at the critical point. It might be possible to generalize the definition of disorder operator for spatial rotation. Another possibility is to consider alternative realizations of the deconfined quantum critical point, such as the one between a quantum spin Hall insulator and a superconductor proposed in arXiv:1811.02583, where both SO(3) and U(1) are realized exactly in the lattice model and their disorder operators can be defined following the standard prescription. We are currently investigating disorder operators in this fermionic model.

---

## Round 3 · Referee Report · Anonymous (Referee 1) · 2022-7-5

Report

I thank the authors for considering the suggested changes to their manuscript. However, regarding two of my previous points, I still request changes:

With regards to my request 2, I don't consider the new Fig. 4 to be useful to understand the quoted values of chi (btw: is this chi/DOF?). The authors can improve on this by simply plotting the relative differences so that one can indeed see the deviations between the data and the red fit line on the scale of the error bars.

With regards to my request 6, the authors have not responded appropriately: I asked them to show in the former Fig. 5 (now Fig. 9) the data for a naive measurement, i.e., without considering the even/odd character. Please include such a measurement as well. Panel 10c is not related to this point.

Requested changes

See report.

  • validity: -
  • significance: -
  • originality: -
  • clarity: -
  • formatting: -
  • grammar: -

Author:  Meng Cheng  on 2022-08-03  [id 2709]

(in reply to Report 1 on 2022-07-05)

We thank referee for his/her useful comments and suggestions, and have implemented the requested changes. Below are our responses to the comments:

Requested changes 1: With regards to my request 2, I don't consider the new Fig. 4 to be useful to understand the quoted values of chi (btw: is this chi/DOF?). The authors can improve on this by simply plotting the relative differences so that one can indeed see the deviations between the data and the red fit line on the scale of the error bars.

Reply 1: Thanks for the suggestion, it is chi-square/DOF. We update the Fig. 4 and add the calculation of of fitting deviations $\Delta_{1(2)}(l)=(X_m(l)-f_{1(2)}(l))/\delta_{X_M(l)}$ with $\delta_{X_M(l)}$ is the error bar of the $X_M(l)$, as show in Fig. 4(b) in the revised manuscript.

Requested changes 2: With regards to my request 6, the authors have not responded appropriately: I asked them to show in the former Fig. 5 (now Fig. 9) the data for a naive measurement, i.e., without considering the even/odd character. Please include such a measurement as well. Panel 10c is not related to this point.

Reply 2: Thanks for the suggestion, we update Fig. 9 in the revised manuscript, i.e., adding the data without considering the even/odd character as Fig. 9(a), as suggested by the referee. Since the data with odd and even boundary of the region $M$ have intrinsic even-odd oscillations, we do not fit but only present them.

---

## Round 3 · Referee Report · Anonymous (Referee 2) · 2022-7-5

Report

I thank the authors for their clarifications and efforts to improve the data analysis.

I still think the discussion of error bars in the main text is too sparse. Results for s(theta) are quoted in the main text with error bars of the order of 2%. But the large deviations between curves in Figure 7 c and 9 c suggest that quantifying the error bar is a nontrivial issue, and it deserves discussion in the main text.

At present, the authors argue that the preferred protocol is safe because it agrees a known result in the J1-J2 case. But in the absence of any direct extraction of an error bar to compare between protocols, (a) how do we know this agreement is not a coincidence, (b) how do we know that the preferred protocol will be safe also in J-Q3 case?

Can the accuracy for each protocol be quantified in some way from the data? The authors give a numerical error bar for one protocol. How is it obtained, and can be it be compared between protocols to give direct comparison?

If the authors cannot directly quantify the error bar then at least there should be some discussion in the main text of the fact that there are nontrivial discrepancies between different protocols.

Apart from this the other issues have been addressed, so when the authors have resolved the above to their satisfaction the paper can be published.
  • validity: -
  • significance: -
  • originality: -
  • clarity: -
  • formatting: -
  • grammar: -

Author:  Meng Cheng  on 2022-08-03  [id 2710]

(in reply to Report 2 on 2022-07-05)

We thank the referee for his/her useful comments and suggestions. Below are responses to the comments:

Comment 1: I still think the discussion of error bars in the main text is too sparse. Results for s(theta) are quoted in the main text with error bars of the order of $2\%$. But the large deviations between curves in Figure 7 c and 9 c suggest that quantifying the error bar is a nontrivial issue, and it deserves discussion in the main text.

Reply 1: We thank the referee for the insightful and professional suggestion and we note that the different curves in Figure 7c and 9c are meant to present the different protocols and we chose the most physical ones from the three and presented the errorbars of $s(\theta)$ within this protocol. In the revised manuscript, we add the sentence as suggested by the referee, that ``we note the quantification of the error bar in $s(\theta)$ is certainly a nontrivial issue, and we use the errorbar of one fitting protocol which gives rise to the correct values of $s(\theta)$ in the J1-J2 case (the other two don't even give the correct values there) but at the same time aware the problem caused among different schemes."

Comment 2: At present, the authors argue that the preferred protocol is safe because it agrees a known result in the J1-J2 case. But in the absence of any direct extraction of an error bar to compare between protocols, (a) how do we know this agreement is not a coincidence, (b) how do we know that the preferred protocol will be safe also in J-Q3 case?

Can the accuracy for each protocol be quantified in some way from the data? The authors give a numerical error bar for one protocol. How is it obtained, and can be it be compared between protocols to give direct comparison?

If the authors cannot directly quantify the error bar then at least there should be some discussion in the main text of the fact that there are nontrivial discrepancies between different protocols.

Reply 2: We thank again the referee for the insightful comment, since different fitting protocol yield different values of $s(\theta)$ and we chose the most physical ones as it gives the correct results in J1-J2 cases, and the other two cannot even agree with such known results. We follow the suggestion of the referee to add the sentence that ``we note the quantification of the error bar in $s(\theta)$ is certainly a nontrivial issue, and we use the errorbar of one fitting protocol which gives rise to the correct values of $s(\theta)$ in the J1-J2 case (the other two don't even give the correct values there) but at the same time aware the problem caused among different schemes."

---

## Round 3 · Author Response

We would like to thank the referees for their careful reading of our paper. The referee 1 gives strong support and he/she ”find that all the general criteria for publication in SciPost Phys. are already met by the manuscript or will be met after my requests for changes are implemented, ···, since this work opens a new perspective and approach to study the DQC scenario, calling for follow-up work from both computational and field-theoretical perspectives.” . Referee 2 fully appreciates ”the significance of this work”, and expect our results ”will stimulate more related research and put steps further of understanding the nature of DQCP. In addition, the method of calculating the disorder operator in QMC itself might also be a useful tool for studying other models and phenomena, and this work would be a good way to broadcast this method”.

Both referees also give valuable suggestions for the improvement of our presentation, which have been incorporated in the main text. We have also added new sections in the Supplemental Materials to address the comments from the referees. With these changes, the narrative of our manuscript has been greatly improved and we really appreciate the help from both respected referees.

---

## Round 3 · List of Changes

1. Responding to the comments of referees, we added Fig.4, Fig.7(b) Fig.9(b) and modified Fig.9(c).

  2. Turned the SM into regular appendices.

  3. Corrected typos and updated references.

---

## Editorial Decision

resubmitted